# A Decoupling Algorithm-Based Technology for Predicting and Regulating the Unbalance of Aircraft Rotor Assembly Considering Manufacturing Errors

**Yingjie Zhao [1], Xiaokai Mu [1,*], Jian Liu [1], Qingchao Sun [1], Ping Zhou [2] and Guozhen Fang [3]**

[1] School of Mechanical Engineering, Dalian University of Technology, Dalian 116023, China; zyj_dlut@mail.dlut.edu.cn (Y.Z.); lj32204223@mail.dlut.edu.cn (J.L.); qingchao@dlut.edu.cn (Q.S.)

[2] Aeroengine Research Institute, Tsinghua University, Beijing 100084, China; zhouping@buaa.edu.cn

[3] Inner Mongolia First Machinery Group Co., Ltd., Baotou 014030, China; 18047226205@163.com

[*] Correspondence: muxiaokai@dlut.edu.cn

**Abstract:** Rotor unbalance is the most important factor affecting the dynamic performance of aircraft engines. The existing unbalance prediction and control methods are insufficient for multi-stage rotors. The post-assembly unbalance of rotors in aircraft engines is a critical factor affecting their dynamic performance. In order to predict and reduce the unbalance of multi-stage rotors after assembly, this paper establishes a measurement model for the center-of-mass offset of aircraft engine rotors through decoupled calculations of the unbalance. Furthermore, it constructs an unbalance prediction model using the spatial transfer mechanism of combined rotor offset centers under the influence of manufacturing errors. Additionally, a method for measuring rotor unbalance during the assembly phase is proposed. The experimental results of the unbalance in multi-stage combined rotor assembly indicate that the degree of agreement between the predicted results and the experimental results is 91.3%, resulting in a reduction in the mean error of 15.3% compared to before the correction. The study also investigates the impact of manufacturing errors on unbalance. This research provides robust support for controlling the unbalance in multi-stage combined rotor assembly.

**Keywords:** aircraft rotor; decoupling algorithm; unbalance; manufacturing errors; prediction; regulation

## 1. Introduction

Aircraft engines are often referred to as the "crown jewel of the industry," characterized by typical features of high temperature, high pressure, and high rotational speed during operation [1]. The high-pressure compressor rotor system, as a core component of aircraft engines, has post-assembly unbalance characteristics that are key factors influencing the performance of the engine's rotor. Unbalance is the primary excitation source for rotor system vibration responses. Therefore, achieving prediction and control of unbalances is especially crucial for enhancing the assembly quality and service performance of aircraft engines [2,3].

Due to the uncertainty in determining the center of mass for each rotor stage and the ambiguous spatial transfer mechanisms during assembly, the prediction of multi-stage rotor unbalance during assembly becomes unreliable [4]. Furthermore, real-time testing of unbalance quantities is challenging during the assembly process, which can lead to final assembly quality not meeting the requirements, subsequently affecting the rotor system's vibration response. Aircraft engine rotors are symmetrical, cylindrical components that rotate. In the high-pressure rotor assembly process, manufacturing errors in various mating surfaces and the assembly phase itself can influence the positions of the center of mass and the rotation axis at each rotor stage, resulting in rotor unbalance. Therefore, research focused on rotor unbalance prediction models that consider manufacturing errors and the

assembly phase is crucial for effectively reducing assembly unbalance and improving rotor operational performance [5,6].

Center-of-mass offset after rotor assembly can lead to rotor unbalance, subsequently inducing vibrations in the entire system. Many experts have studied the impact of post-assembly center-of-mass offset-induced rotor unbalance on engine vibrations. Qing et al. [7] established a model for the breathing behavior of cracks with mass eccentricity and conducted a specific study on the torsional effects of unbalance orientation angles. Liu [8] proposes a method to minimize stage-by-stage initial unbalance in aero engine assembly of multi-stage rotors based on the connective assembly model. The analysis includes the propagation of mass eccentric deviation in the assembly, and the effectiveness of the proposed method is verified through the assembly of multi-stage rotors using the optimal assembly strategy. Wang's [9] study focuses on the dual-rotor system supported by dual bearings, where the Riccati transfer matrix method with good numerical stability is used to establish the model of the magnetic suspended dual-rotor system unbalance response. Finally, the dynamic characteristics of the unbalance response are investigated. M.B. et al. [10,11] employed the transfer matrix method to analytically derive the influence coefficients for a rotor-bearing system with both mass unbalance and bow. They identified distributed unbalance through the investigation of a polynomial curve representing the eccentricity distribution. Furthermore, the unbalance distribution is estimated by analyzing vibration responses measured at speeds below the balancing speed. Wang, LK et al. [12] investigated the impact of unbalance location on the critical speed and vibration characteristics of a double-overhung rotor. Moreover, numerous scholars have investigated techniques for post-assembly adjustments of the geometric attributes and unbalance properties of aviation engines. The prediction model of rotor unbalance often encounters difficulties in dealing with many stages of a rotor. Mu and Sun et al. [13,14] studied an assembly accuracy prediction method for aviation engine rotors that takes into account manufacturing errors and assembly deformations. Additionally, a spatial transfer model for offsetting the center of mass was proposed, enabling control over assembly accuracy and unbalance. Zhang et al. [15] proposed a model-based rotor-balancing method, followed by the utilization of the differential evolution algorithm to obtain the optimization solution. Zhang et al. [16] presented a method for representing geometric errors using Non-Uniform Rational B-Spline (NURBS) surfaces. This approach enables the incorporation of geometric errors in the virtual modeling of mechanical assemblies. Li et al. [17] proposed a datum error elimination method that takes into account the independence of geometric characteristics from the measurement datum. This method enhances the accuracy of the rotor characteristic matrix and assembly model. Zhu Z et al. [18] conducted an analysis of the impact resulting from variations in assembly sequences on the geometric deviation of critical features within an assembly. This information can be used to identify specific assembly sequences for evaluation, potentially serving as valuable evidence for optimizing the assembly process planning. Sun et al. [19,20] established a transfer model for mass eccentricity and introduced a novel method for the propagation and control of unbalance during the assembly process. Many experts and scholars have conducted corresponding research on the prediction and real-time testing of unbalance during the assembly process of multi-stage rotors. Li R et al. [21] constructed a measurement model for the centroid deviation of aero-engine rotors by decoupling unbalance. This model reveals the spatial transmission mechanism of centroid deviation in the assembled rotors, taking into account the influence of machining errors. Yue Chen et al. [22] proposed a method for optimizing the unbalance of a multi-stage rotor during assembly. In contrast to traditional methods that rely on static features, Lan L et al. [23] proposed a method that combines dynamic features with support vector machines (SVM) for the precise detection and classification of rotor unbalance faults. Sudhakar et al. [24] introduced a comprehensive methodology for fault identification that aims to minimize errors even when there are fewer measured vibrations available in the current study. The measurement method for centroid deviation encompasses techniques

such as the weighing method [25–27], multiline torsional pendulum method [28,29], and modal identification method [30,31], among others.

During the assembly of aircraft engine rotors, the axis of rotation changes as the assembly progresses, causing errors in the center-of-mass deviation from the rotation axis at each stage to vary in real time. This makes accurate prediction of unbalance quantities challenging. Additionally, the difficulties in conducting real-time unbalance testing during the assembly process result in significant deviations in the final unbalance quantities, failing to meet the assembly requirements.

To address this issue, this paper focuses on the measurement of unbalance during the assembly process of multi-stage rotors and on accurate prediction. The following tasks have been undertaken: Establishment of a measurement model for the centroid deviation of engine rotors and a spatial transfer model during the assembly phase. Based on these models, a predictive model for the unbalance of multi-stage rotors has been constructed and a method for measuring rotor unbalance during the intermediate assembly stages introduced. This method involves the correction of original prediction results using intermediate test data and optimization of rotor unbalance through assembly-phase adjustments.

## 2. Rotor Unbalance Prediction Model

### 2.1. Decoupling Algorithm of Rotor Unbalance

To establish a centroid transfer model during the rotor assembly process, it is necessary to decouple the unbalance of individual stages to obtain the centroid positions of each stage.

As shown in Figure 1, during the unbalance testing of aircraft engine rotors, two unbalance correction planes are established, namely Correction Plane 1 and Correction Plane 2. The unbalance measurement on these two planes is denoted as $\omega_1$ and $\omega_2$, respectively. The overall unbalance of the aircraft engine, denoted as $\omega$, is the vector sum of the unbalances on these two correction planes. Due to the non-uniform shape of the rotor and manufacturing irregularities, the actual center-of-mass position must be obtained through decoupling from the unbalance measurements. It can be assumed that two unbalance mass centers exist on the two correction planes, respectively. Their positions are defined in a coordinate system with the front face of the rotor serving as the reference plane, as described by Equation (1).

$$\left(x_1, \frac{w_1 \cos \varphi_1}{m}, \frac{w_1 \sin \varphi_1}{m}\right); \left(x_2, \frac{w_2 \cos \varphi_2}{m}, \frac{w_2 \sin \varphi_2}{m}\right) \tag{1}$$

Herein, $x_1$ and $x_2$ represent the distances from the two correction planes to the coordinate origin, while $\varphi_1$ and $\varphi_2$ denote the phases of unbalance $\omega_1$ and $\omega_2$, respectively. Furthermore, '$m$' denotes a mass unit conversion factor, which is equal to the rotor's mass in the equation.

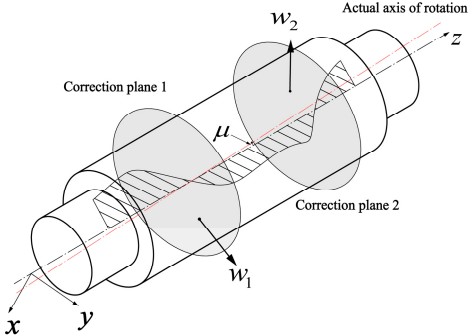

**Figure 1.** Decoupling of Dual Correction Plane Balancing for Aeroengine Rotors.

The position of the rotor's centroid in the single-body rotating coordinate system is expressed by the equation:

$$\left(x_1 + \frac{w_2}{w_1 - w_2}(x_2 - x_1), \frac{w_1 \cos \varphi_1 + w_2 \cos \varphi_2}{m}, \frac{w_1 \sin \varphi_1 + w_2 \sin \varphi_2}{m}\right) \tag{2}$$

The offset of the centroid relative to the actual rotation axis of the rotor is expressed as follows:

$$\mu = \frac{\sqrt{w_1^2 + w_2^2 + 2w_1 w_2 \cos(\varphi_1 - \varphi_2)}}{m} \tag{3}$$

The phase of this offset centroid can be expressed as:

$$\varphi = \frac{w_1 \sin \varphi_1 + w_2 \sin \varphi_2}{w_1 \cos \varphi_1 + w_2 \cos \varphi_2} \tag{4}$$

The quantities involved in Equations (1)–(4) include 'm', which represents the rotor's mass and can be measured using a mass scale. The rotor's two unbalance quantities, $\omega_1$ and $\omega_2$, along with their phases $\varphi_1$ and $\varphi_2$, can be obtained through testing on a vertical balancing machine. Through the decoupling calculations outlined above, the position of the rotor's centroid in the single-body rotating coordinate system, denoted as '$c_i$', can be determined. Subsequently, the centroid offset and its phase offset can be further calculated.

*2.2. Prediction Model for Multi-Stage Rotor Unbalance*

The high-pressure compressor rotor system is assembled from multiple stages of rotors, and the errors in rotor centroids relative to the axis of rotation change as the assembly process progresses, consequently affecting the final unbalance. Therefore, constructing a centroid transfer model for multi-stage rotor assembly and determining the ultimate axis of rotation are crucial for accurately predicting the unbalance quantities.

The position of a single-stage rotor in its single-body rotating coordinate system is denoted as $c_i$. Following the actual assembly sequence of the aeroengine rotor, the centroids of each rotor stage are represented as $c_1, c_2, c_3, \ldots c_n$ (where 'n' is the number of stages in the aeroengine rotor). The centroid deviations for each rotor stage are $u_1, u_2, u_3, \ldots u_n$, and the corresponding phases of these centroid deviations, which are the same as those for static unbalance, are denoted as $\varphi_1, \varphi_2, \varphi_3 \ldots \varphi_n$. The total centroid deviation for the assembled rotor is '$u_m$'.

Assuming that the fit between parts is ideal with no manufacturing errors, and the actual rotation axis after assembly coincides with the ideal rotation axis, the transmission relationship of the centroids of the rotor at different stages is only related to the position of the centroid of each single-stage rotor in its own single-body rotating coordinate system. However, due to manufacturing errors between the mating surfaces of parts, errors accumulate and transfer between different mating surfaces along the assembly path as the assembly process progresses. Under different assembly phases, the rotor generates different rotation axes, leading to changes in the offset of the centroids of each rotor stage relative to the rotation axis. Therefore, the machining error of the rotor is a factor affecting the actual rotation axis of the rotor after assembly, while the position of the rotor's centroid determines the distance of the centroid deviation from the rotation axis after assembly. These factors are coupled with each other. To understand the mutual influence between them, it is necessary to establish the transformation relationships between the single-body rotating coordinate system, single-body coordinate system, assembly coordinate system, and assembly rotating coordinate system when assembling multi-stage rotors. The schematic diagram depicting the relationships between these coordinate systems is shown in Figure 2.

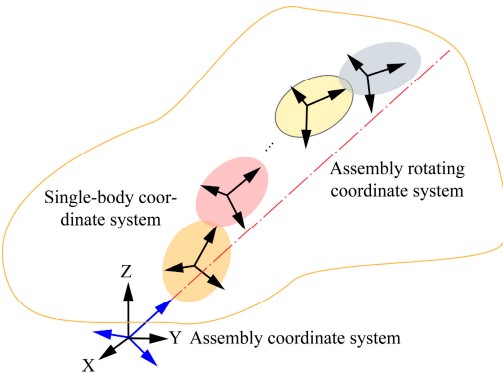

**Figure 2.** Schematic diagram illustrating the relationships between the coordinate system transformations.

The position matrices of the centroids of each rotor stage in their respective single-body rotating coordinate systems can be represented as follows:

$$c_i = [x_i, y_i, z_i] \tag{5}$$

where $x_i$, $y_i$, $z_i$ can be calculated through the decoupling calculations based on Equation (2).

The position matrix of the centroid of a single-stage rotor in a single-body coordinate system with its front face as the coordinate base can be represented as follows:

$$c_i^s = [x_i^s, y_i^s, z_i^s] \tag{6}$$

where $x_i^s$, $y_i^s$, $z_i^s$ represent the centroid coordinates of the i-th rotor along the x, y, and z axes in its single-body coordinate system. The transformation relationship between Equations (5) and (6) is represented as

$$\begin{pmatrix} x_i^s \\ y_i^s \\ z_i^s \end{pmatrix} = \begin{pmatrix} 1 & 0 & \theta_y \\ 0 & 1 & -\theta_x \\ -\theta_y & \theta_x & 1 \end{pmatrix} \begin{pmatrix} x_i \\ y_i \\ z_i \end{pmatrix} \tag{7}$$

where $\theta_x$ and $\theta_y$ represent the machining errors of the front face of the parts.

The position matrix of the centroid of the *i*-th rotor stage in the assembly coordinate system can be obtained from its position matrix in the single-body coordinate system, as in Equation (8).

$$c_i^a = (\prod_1^{i-1} R_k) \cdot c_i^s \tag{8}$$

where $R_k$ is the spatial pose representation matrix of the *k*-th rotor stage. During assembly, each rotor stage can be rotated around its *z*-axis to adjust the assembly phase. Therefore, $R_k$ contains factors related to phase rotation. According to the theory of homogeneous transformations, $R_k$ can be represented as given in the equation:

$$R_k = \begin{bmatrix} cos\alpha_k & -sin\alpha_k & \Delta\theta_y & 0 \\ sin\alpha_k & cos\alpha_k & -\Delta\theta_x & 0 \\ -\Delta\theta_y cos\alpha_k + \Delta\theta_x sin\alpha_k & \Delta\theta_y sin\alpha_k + \Delta\theta_x cos\alpha_k & 1 & h \\ 0 & 0 & 0 & 1 \end{bmatrix} \tag{9}$$

Here, $\alpha_k$ represents the phase value for the assembly of the respective rotor stage, $\Delta\theta_x$ and $\Delta\theta_y$ denote the differences in angular offsets around the *x*-axis and *y*-axis, respectively, for the fitting planes of the front and back faces of the rotor stage. Additionally, '*h*' represents the height of the respective rotor stage.

Therefore, the end face's error plane can be fitted using the least squares method to obtain the expression for the machining surface, and the radial runout data can be fitted to

determine the center of the error surface. The expression for the actual assembly surface of the *i*-th rotor stage, obtained through fitting, is given by

$$A_i x + B_i y + C_i z + D_i = 0 \tag{10}$$

Here, $A_i$, $B_i$, $C_i$, and $D_i$ represent the coefficients of the *i*-th rotor stage's end face equation. Due to the high machining accuracy of the rotor contact surface, the deviation error of the end face caused by machining errors is extremely small. According to reference [32], the deflection error of the *i*-th rotor stage's end face around the *x* and *y* axes is given by

$$\theta_{xi} = -\frac{B_i}{C_i}, \theta_{yi} = -\frac{A_i}{C_i} \tag{11}$$

After obtaining the positions of centroids for each stage of the rotor in the assembly coordinate system, it is also necessary to determine the rotor's axis of rotation. It is known that the rotation axis passes through the origin *O* (0, 0, 0) of the assembly coordinate system and the center $O_n^A = \left( x_n^A, y_n^A, z_n^A \right)$ of the upper surface of the last-stage rotor. The coordinates of the center of the upper surface can be obtained using the Equation (12). The schematic representation of the centroids of each rotor stage and the rotation axis during assembly are shown in Figure 3.

$$O_n^A = \left( \prod_1^{n-1} R_k \right) \cdot P_n^A \tag{12}$$

Here, $O_n^A$ represents the center of the upper surface of the last-stage rotor in the assembly coordinate system, and $P_n^A$ represents the center of the upper surface of the last-stage rotor in its individual coordinate system.

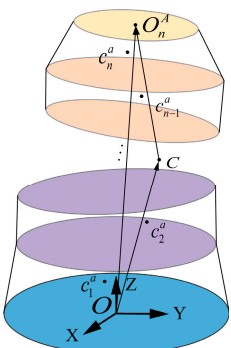

**Figure 3.** Schematic diagram illustrating the centroid position and axis of rotation after the assembly of a multi-stage rotor.

The vector representing the rotation axis can be expressed as follows:

$$n = \left( x_n^A, y_n^A, z_n^A \right) \tag{13}$$

The offset of the centroids of each stage of the assembled aircraft engine rotor relative to the rotation axis is as follows

$$\mu_i = \frac{\left| \vec{C_i O_n^A} \times \vec{n} \right|}{|n|} \tag{14}$$

The phase of the centroid offset for each stage of the rotor can be expressed as follows, based on the equation

$$\varphi_i = arctan \frac{y_i^R}{x_i^R} \tag{15}$$

It can be easily obtained that the decomposed unbalance quantity of the aircraft engine rotor is as follows, based on the equation

$$w_{ix} = \sum \mu_i m_i cos\varphi_i, w_{iy} = \sum \mu_i m_i \sin \varphi_i \tag{16}$$

The unbalanced quantity of the assembled aircraft engine rotor is given by

$$w = \sqrt{w_{ix}^2 + w_{iy}^2} \tag{17}$$

As evident from the above, the position of the center of mass for the *i*th-stage rotor is influenced by the machining errors of the preceding $i - 1$ stages, resulting in both rotation and translation transformations in its individual coordinate system. Additionally, with the assembly of each rotor stage, there is a change in the direction of the actual rotation axis. Once we obtain the positions of the centers of mass of each rotor stage in the assembly coordinate system and the direction vectors of the rotation axes, we can calculate the center of mass offsets of each rotor stage relative to the rotation axis. Multiplying these offsets by the masses of each rotor stage yields the unbalanced quantity of the assembled rotor. Furthermore, by adjusting the assembly phases of each rotor stage, we can fine-tune the total rotor unbalance quantity.

### 3. Model Correction Based on Experimental Values as Inputs

*3.1. Unbalance Measurement Method during the Assembly Synchronization Process*

During the assembly process of multi-stage rotors, unfinished rotors, due to the absence of bolt connections or the lack of corresponding balancing fixtures, cannot be tested for unbalance on the unbalance machine at the intermediate assembly stage. As a result, it is impossible to determine the real-time unbalance status of the already assembled rotors, which is detrimental to the subsequent rotor assembly process. Therefore, a precision-based method for measuring rotor unbalance is proposed. This method involves precision testing to calculate the rotor's unbalance status, providing valuable information for the subsequent assembly. Additionally, it saves both the effort and economic costs associated with unbalance testing.

Taking the assembly of the $i - 1$ rotor component and the *i*-th rotor as an example, through deduction it can be determined that we have already measured the unbalance of the $i - 1$ level rotor component, denoted as $\omega_{i-1}$. The centroid position of the i-1 rotor component in the assembly coordinate system is known as $c_{i-1}^a = \left[ x_{i-1}^a, y_{i-1}^a, z_{i-1}^a \right]$. The position of the *i*-th level rotor in its own body-revolving coordinate system is given by $c_i = [x_i, y_i, z_i]$. We first transform the centroid position coordinates of the *i*-th rotor component to a coordinate system with the non-assembly face (the rear end face of the rotor) as the coordinate base. The coordinate transformation is described by the equation

$$c_i^* = \theta_e c_i^a \tag{18}$$

Here, $c_i^* = \left[ x_i^*, y_i^*, z_i^* \right]$ represents the centroid position of the *i*-th level rotor in a coordinate system with the non-assembly face as the coordinate base. $\theta_e$ is the machining error matrix for the non-assembly face, and its form is the same as the one presented in Equation (7).

After assembling the *i*-th rotor component with the $i - 1$-th rotor component, the position of the *i*-th rotor's center of mass in the assembly coordinate system can be determined by the equation

$$\begin{pmatrix} x_i^R \\ y_i^R \\ z_i^R \\ 1 \end{pmatrix} = \begin{bmatrix} 1 & 0 & \theta y' - \theta y & dx' - dx \\ 0 & 1 & \theta x - \theta x' & dy' - dy \\ \theta y - \theta y' & \theta x' - \theta x & 1 & \sum h_i \\ 0 & 0 & 0 & 1 \end{bmatrix} \begin{pmatrix} x_i^* \\ y_i^* \\ z_i^* \\ 1 \end{pmatrix} \tag{19}$$

where $c_i^a = \begin{bmatrix} x_i^a, y_i^a, z_i^a \end{bmatrix}$ represents the center of mass of the $i$-th rotor component in the assembly coordinate system, $\theta x$, $\theta y$, $\theta x'$ and $\theta y'$ represent the deflection values of the upper and lower surfaces of the i-th assembly, and $dx$, $dy$, $dx'$ and $dy'$ represent the eccentricity values of the centers of the upper and lower surfaces. Once the positions of the center of mass for the previous $i - 1$ rotor components and the $i$-th rotor are determined, the unbalance of the previous $i$ rotor components can be calculated, as discussed in the preceding section.

### 3.2. Method of Model Correction with Intermediate Test Measurements as Input

In multi-stage rotor unbalance prediction, the prediction error increases rapidly with the number of rotor stages, rendering existing prediction models ineffective and inadequate. This paper employs a model correction method that uses actual measurements as input to refine the unbalance prediction model for multi-stage rotors. The schematic diagram of the correction method is illustrated in Figure 4.

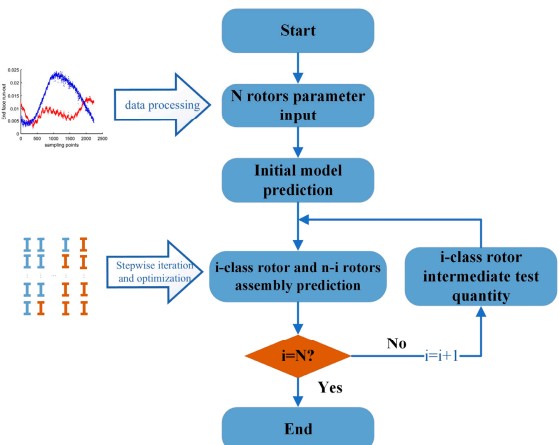

**Figure 4.** Schematic Diagram of the Correction Principle for the Unbalance Optimization Model in N-Stage Rotor Assembly.

In the example presented in this paper, the assembly of the seven-stage rotor is initially predicted using the initial prediction model. This yields the initial assembly phases for the seven-stage rotor. Then, the test measurements for the $i$-stage rotor component are input into the prediction model, and adjustments are made to the initial phases of the subsequent $n - i$ stages. This process is repeated iteratively.

## 4. Experimental Design

### 4.1. Measurement of Rotor Unbalance

The experimental rotor structure in this study is illustrated in the diagram. The experimental rotor consists of seven components, each of which is labeled in Figure 5 with their respective names, including stages 1 to 6 and the drum shaft. Referring to the decoupling method for single-stage rotors described in Section 2, each rotor stage can be tested using the horizontal balancing machine depicted in Figure 6, which provides unbalance measurements and phase information on two correction planes. The unbalance test results for the 7th rotor stage are presented in Table 1.

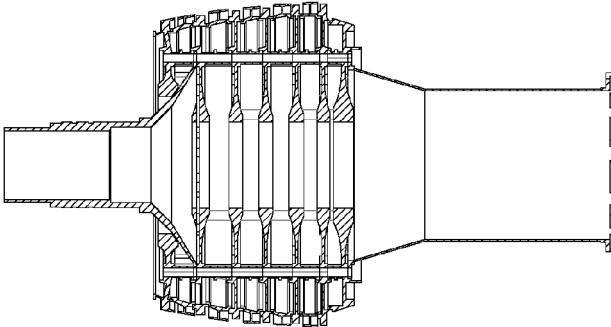

**Figure 5.** Schematic diagram of the experimental rotor's structure.

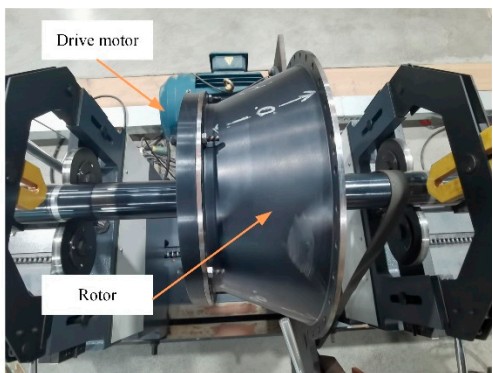

**Figure 6.** Schematic diagram of horizontal balancing machine.

**Table 1.** Unbalance test results of each rotor stage.

| Rotors | $\omega_1$(g.mm) | $\varphi_1$(°) | $\omega_2$(g.mm) | $\varphi_2$(°) |
|--------|------------------|----------------|------------------|----------------|
| 1 | 169 | 84 | 147 | 256 |
| 2 | 273 | 225 | 98 | 72 |
| 3 | 229 | 200 | 77 | 230 |
| 4 | 78 | 215 | 165 | 188 |
| 5 | 164 | 305 | 118 | 322 |
| 6 | 53 | 77 | 284 | 174 |
| 7 | 120 | 95 | 254 | 33 |

According to Equation (2), the centroid of a single-stage rotor can be decoupled by analyzing the unbalance measurements on the two correction planes. This provides the position of the single-stage rotor's centroid in its individual rotary coordinate system. Then, using Equation (7), you can determine the positions of each rotor stage's centroid in their respective individual coordinate systems. Furthermore, based on Equation (8), you can calculate the positions of each rotor stage's centroid in the assembly coordinate system. The offset of each centroid relative to the rotation axis can be obtained using Equation (14). The total unbalance quantity of the assembly can then be computed using Equation (17). By adjusting the assembly phase using the original prediction model, you can optimize the unbalance quantity for the seven-stage rotor under these manufacturing errors. Additionally, after adjusting the initial prediction model based on intermediate test measurements, the adjusted phase results are shown in Figure 7, illustrating the changes in the size and position of the offset centroids before and after adjustment.

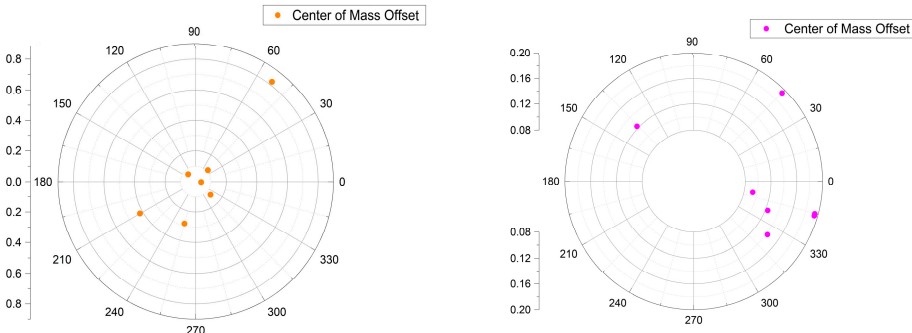

**Figure 7.** Schematic Diagram of Center of Mass Offset in Before and After Correction.

As shown in Figure 6, this paper used a horizontal balancing machine to measure the unbalance of the assembled rotor. According to the multi-stage rotor unbalance prediction model proposed in Section 2, the optimal assembly angles for each stage of the rotor were determined. According to the proposed approach in this paper, the unbalance predictions before and after model correction, as well as the phase of assembly for each stage of the rotor, are shown in Figure 8.

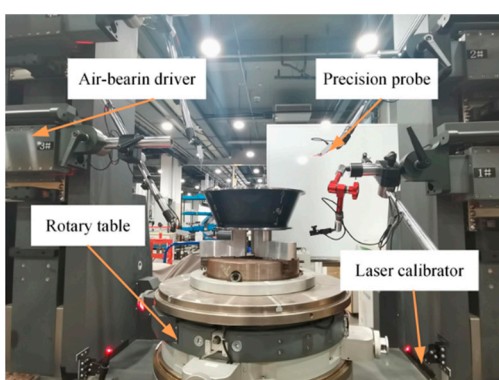

**Figure 8.** Precise air-bearing turntable.

Simultaneously, the effectiveness of the model is demonstrated through a set of control groups. The predicted results of the three groups were 900, 580 and 520 g.mm, respectively. From Figure 9, it can be observed that, when using the proposed model to guide assembly, the unbalance levels before and after correction decreased by 35.5% and 42.2% relative to the control experiments, respectively. The model resulted in a 15.8% reduction in unbalance levels after correction compared to before. According to the three different assembly phases, the experimental results of the unbalance quantity obtained are 990, 765 and 570 g.mm, respectively. The experimental results of the seven-stage rotor assembly show that the degree of agreement between the predicted results and the experimental results is 91.3%, resulting in a reduction in the mean error of 15.3% compared to before the correction.

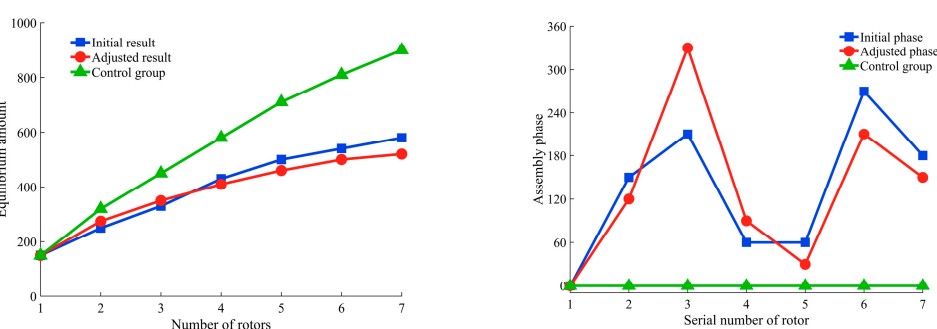

**Figure 9.** Comparison of experimental results for unbalance levels and phase.

### 4.2. Measurement of Rotor Manufacturing Errors

According to the spatial transfer model of the rotor's center of mass proposed in this paper during the assembly process, it is necessary to measure the manufacturing errors in the radial and axial directions of each rotor stage. In this study, a precision air-bearing turntable, as shown in Figure 9, was used to measure manufacturing errors. The turntable is equipped with four TESA-GT31 sensors, which measure the axial and radial surface machining errors at the front and rear ends of the rotor. The measurement accuracy is 0.1 um, the repeatability is 0.1 um, and the angular offset accuracy is 0.3″. The measurement results of manufacturing errors for each rotor are presented in Table 2.

**Table 2.** Measurement results of manufacturing errors for each rotor stage.

| Rotors | Eccentricity Error | | Tilt Error | |
|:---:|:---:|:---:|:---:|:---:|
| | $d_x$(mm) | $d_y$(mm) | $\theta_x$(e − 5) | $\theta_y$(e − 5) |
| 1 | 0.0071 | −0.0080 | 0.9258 | 3.0230 |
| 2 | −0.0163 | 0.0012 | 1.8970 | 3.6620 |
| 3 | 0.0020 | 0.0113 | 0.6221 | 2.2835 |
| 4 | 0.0125 | −0.0015 | 1.4430 | 0.0398 |
| 5 | −0.0065 | −0.0027 | 1.3600 | 0.3945 |
| 6 | −0.0031 | 0.0037 | 0.3258 | 1.5660 |
| 7 | 0.0094 | −0.0045 | 0.7441 | 1.8358 |

## 5. Discussion

### 5.1. The Relationship between Unbalance and Centroid Concentricity

To study the mutual influence between centroid concentricity and the unbalance of the rotor, we provide a brief analysis as follows. The position matrix of the center of mass of each rotor stage in the assembly coordinate system can be expressed as $c_i^a = \left[ x_i^a, y_i^a, z_i^a \right]$. Consequently, the positions of the center of mass of each rotor stage in the assembly rotating coordinate system can be represented as $c_i^R = R_n \cdot c_i^a$, where

$$R_n = \begin{bmatrix} cos\beta & 0 & -sin\beta \\ sin\alpha sin\beta & cos\beta & sin\alpha cos\beta \\ cos\alpha sin\beta & -sin\alpha & cos\alpha cos\beta \end{bmatrix} \tag{20}$$

Describing the concentricity error of the centroid distribution by using the actual rotation axis as the reference axis and considering the minimum cylinder radius d that encompasses all centroids,

$$d = \max(u_1, u_2, u_3 \ldots u_n) \tag{21}$$

The concentricity error of the centroid distribution represents the range of the centroid positions of individual components in the assembly. During the operation of the rotor, since the centroids are not aligned in a straight line, it can introduce additional bending moments to the mating surfaces of the components. Rotor assemblies with dispersed centroid distributions can also result in significant additional unModal balancing of flexible rotors with bow and distributed unbalance if deformation occurs during operation. Therefore, while predicting the unbalance of the assembly, it is essential to consider the distribution of the centroids of individual components.

We observed that controlling the assembly phase to minimize rotor unbalance often results in a larger eccentricity error for the centroids of each rotor stage. Conversely, controlling only the eccentricity error does not guarantee the rotor's unbalance after assembly. Taking the example of a seven-stage rotor assembly, when the minimum value for total unbalance is achieved at 730, the eccentricity error for each rotor stage is 0.14. When controlling the eccentricity error to a minimum of 0.06, the total unbalance increases to 1160. As shown in Figure 10, as the centroid eccentricity varies from 0.06 to 0.30, the rotor's optimal unbalance first decreases and then increases. The rotor's unbalance is minimized when the eccentricity error is around 0.14 mm.

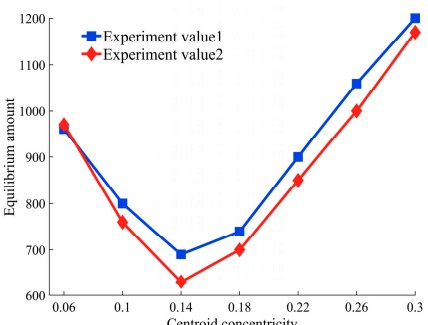

**Figure 10.** The relationship between the optimal unbalance and the centroid concentricity for the rotor.

### 5.2. The Impact of Manufacturing Errors on Unbalance

To study the influence of manufacturing errors on the minimum unbalance quantity, rotor masses were set at 3 kg, 4 kg, and 3.5 kg, with rotor heights of 40 mm, 45 mm, and 30 mm. Rotor eccentricity errors were set as shown in the table, and tilt errors were set at 0.5′, 1.0′, 1.5′, 2.0′, 2.5′, 3.0′, and 3.5′. Changing the assembly angle can optimize the total unbalance quantity and concentricity of the assembly. The relationship between tilt error and minimum unbalance quantity was plotted using numerical simulation results, as shown in Figure 11. The results for rotor eccentricity errors are presented in Table 1. When the rotor eccentricity errors were $d_x = 0.003$ and $d_y = 0.004$, the corresponding assembly rotor unbalance quantities for different tilt errors were 304, 268, 231, 245, 432, 578, and 729. The results indicate that as the tilt error increases, the unbalance quantity of the assembled rotor first decreases and then increases. The initial decrease is due to the overall centroid offset caused by tilt errors being offset by the centroid offset caused by eccentricity errors. However, as the tilt error continues to increase, the additional eccentricity caused by tilt errors gradually dominates, causing the centroids to move away from the axis, resulting in an increase in the total unbalance quantity of the assembled rotor.

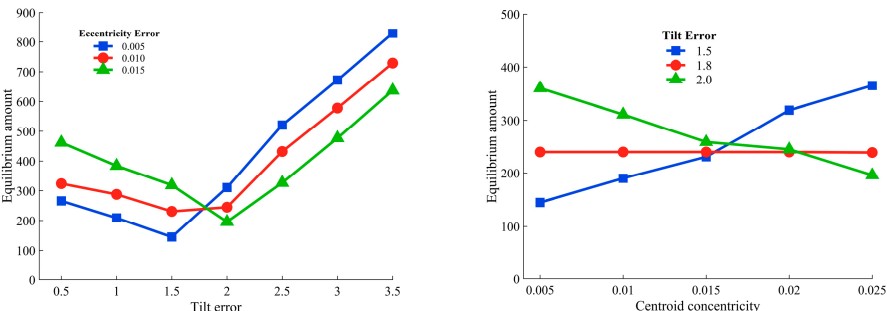

**Figure 11.** The relationship between the optimal unbalance quantity and tilt error.

To study the influence of manufacturing errors on the concentricity when achieving the minimum unbalance quantity, the curve shown in Figure 12 was plotted. As the tilt error increases, the concentricity error when reaching the minimum unbalance quantity shows a trend of first decreasing and then increasing. Reasonable control of tilt errors during the manufacturing phase has a positive impact on the optimization results of unbalance quantity and concentricity after assembly. At the same time, due to the less pronounced effect of eccentricity error, as shown in Figure 12, when the tilt error is 2″, the concentricity error when achieving the minimum unbalance quantity remains around 0.06 as the eccentricity error increases from 0.005 to 0.015.

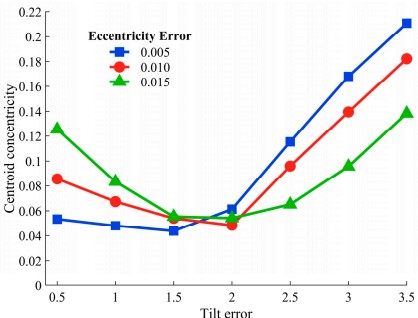

**Figure 12.** The influence of tilt error on the concentricity when achieving the minimum unbalance quantity.

## 6. Conclusions

This paper establishes a measurement model for the offset of the center of mass of an aircraft engine rotor by decoupling the unbalance of a single-stage rotor. It also constructs an initial prediction model for unbalance levels using the spatial transfer mechanism of combined rotor offset centers influenced by manufacturing errors and assembly phase. Additionally, it proposes a method for measuring rotor unbalance during the assembly-incomplete stage and refines the initial prediction model using intermediate measurement data. Results from a seven-stage rotor assembly experiment demonstrate that, under the control of the initial prediction model, the unbalance levels decreased by 35.5% compared to the control group.

(1) After model correction, the assembly unbalance decreased by 42.2% compared to the control group. The unbalance levels after model correction reduced by 15.8% compared to before correction, confirming the effectiveness of the model adjustment. The theoretical values after model correction matched experimental values by 91.3%, and the average error is reduced by 15.3% compared to before correction.

(2) The relationship between the concentricity of the center of mass and unbalance levels was explored. As concentricity decreased, assembly unbalance initially decreased and then increased. During rotor assembly, both factors should be reasonably considered.

(3) The impact of manufacturing errors on unbalance levels and concentricity was studied. With increasing tilt error, unbalance initially decreased and then increased. When tilt error falls within a reasonable range, eccentricity error has minimal impact on unbalance. As tilt error increases, the concentricity at the point of optimal unbalance initially decreases and then increases. As eccentricity error increases from 0.005 to 0.015, the concentricity at the point of optimal unbalance remains around 0.06. To simultaneously control rotor concentricity while achieving optimal unbalance, particular attention should be paid to tilt errors during rotor machining and subsequent assembly. These research findings can effectively support the quality adjustment of multi-stage rotor assemblies.

**Author Contributions:** Conceptualization, X.M., Q.S. and Y.Z.; methodology, X.M., Q.S. and Y.Z.; software, Y.Z. and J.L.; validation, Y.Z., J.L. and P.Z.; investigation, P.Z. and G.F.; data curation, Y.Z. and J.L.; writing—original draft preparation, Y.Z.; writing—review and editing, Y.Z. and X.M.; supervision, Q.S., X.M. and Y.Z.; project administration, X.M. and Q.S.; and funding acquisition, X.M., P.Z. and G.F. All authors have read and agreed to the published version of the manuscript.

**Funding:** This research received no external funding.

**Data Availability Statement:** For the sake of data security, we regret that we cannot disclose the original data.

**Acknowledgments:** This work was supported by the National Natural Science Foundation of China (52375483,52005081); Natural Science Foundation of Liaoning Province of China (2023-MS-104).

**Conflicts of Interest:** The authors declare no conflict of interest.

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
