# Peer review of "A Decoupling Algorithm-Based Technology for Predicting and Regulating the Unbalance of Aircraft Rotor Assembly Considering Manufacturing Errors"

_machines, doi:10.3390/machines11100970_

Round 1

Reviewer 1 Report

This paper discusses a predictive method for the unbalance measurement of an aircraft engine rotor. The decoupling calculation of the rotor's center of mass is of significant importance in accurately predicting the unbalance measurement. It is effective to use intermediate test data as input to adjust the initial prediction results. The paper is well-structured and exhibits a sufficient level of innovation. I would suggest its acceptance after a minor revision to address the following issues:

1.     Some vectors in certain formulas should be in bold.

2.     Improve the quality of the images, especially for Figures 4 and 7.

3.     For the concentricity described in the discussion, it's necessary to add illustrative images.

4.     Do equations (8) and (9) simultaneously consider both manufacturing errors and assembly phase? What is the minimum angle that can be adjusted for the assembly phase? and does the magnitude of the minimum adjustment angle have a significant impact on the unbalance?

5.     Manufacturing errors can be regarded as parametric uncertainties. Please discuss this point by referencing to relevant literature on uncertainty handling in rotor systems.

6.     Please provide original diagrams as the figures before fig. 9 in the PDF are blur and have low resolution.

7.     Can you present the effects of these errors on rotordynamics, such as responses?

Reviewer 2 Report

Dear Authors

The submitted manuscript is dealing with a measurement model on the offset of mass center of an aircraft engine rotor. The paper is in a good structure, good language and comprehensive approach.

It owns high merit in terms of scientific aspect and it will be helpful for both academy and industrial partners. It is descriptive in fundamentals of the topic. In general, the paper is interesting but requires some corrections for improvement.

-       Please reproduce the figures with  higher resolution and better quality,

-       Page 13, please explain how to set the tilt errors and why these values?

-       Wherever presenting equations, please cite a reference for that unless it has been developed by the Authors.

-       The errors reported on Table 2, how were they computed? Based on any formulations? If so, please indicate.

-       Please respect the same decimals all over the paper.

Best regards,

The Reviewer
